# Exploring Cognitive Stimulation as a Therapy for the Prevention of Delirium in a Hospital Setting: A Narrative Review

**DOI:** 10.3390/bs15020186

**Published:** 2025-02-11

**Authors:** Emman Fatima, Ian Hill, Noah Dover, Hina Faisal

**Affiliations:** 1School of Medicine, Creighton University, Omaha, NE 68178, USA; emmanfatima@creighton.edu (E.F.); ianhill@creighton.edu (I.H.); 2Texas A&M School of Engineering Medicine and Houston Methodist, Houston, TX 77030, USA; nbdover@tamu.edu; 3Center for Critical Care, Houston Methodist, Houston, TX 77030, USA; 4Department of Surgery, Houston Methodist, Houston, TX 77030, USA

**Keywords:** delirium, cognitive stimulation, cognitive training, cognitive rehabilitation, cognitive prehabilitation, geriatric, prevention

## Abstract

Delirium is a highly prevalent and costly neuropsychiatric condition that affects up to 87% of critically ill hospitalized patients. It impacts various cognitive domains, including attention, memory, perception, and motor functions, with symptoms such as behavioral changes, hallucinations, slurred speech, visual impairments, and fatigue. Non-pharmacological interventions have been shown to reduce the incidence and duration of delirium, with strategies like reality orientation, cognitive stimulation, family support, and physical support. However, the scalability of these interventions in hospital settings is limited by resource constraints, low patient engagement, and the complexity of delivery. Digital technology-based cognitive stimulation offers a potential solution to these barriers, though evidence supporting its effectiveness is currently insufficient. This narrative literature review explores both traditional and novel digital technology-based cognitive stimulation techniques for the prevention and management of delirium in acute hospital settings.

## 1. Introduction

Delirium is a common, serious, and costly neuropsychiatric syndrome with a multifactorial etiology, often under-recognized and associated with significant burden in critically ill patients, particularly in the intensive care unit (ICU) ([29]; [32]). The incidence of delirium in ICU settings varies, with reports indicating up to 87% prevalence in older critically ill patients ([1]). Delirium affects multiple cognitive domains, including attention, disorganized thinking, vigilance, memory, and executive function ([22]). Symptoms commonly include inattention, disorientation, hallucinations, slurred speech, visual disturbances, and fatigue ([14]).

Major risk factors for ICU delirium include non-modifiable predisposing factors (e.g., age, prior cognitive impairment, dementia, pre-illness frailty) and acute precipitating factors related to the illness requiring hospitalization ([1]; [30]; [54]). There is no consensus regarding the treatment of delirium, due to an incomplete understanding of its pathogenesis. Although there is no FDA-approved pharmacotherapy to prevent or treat delirium ([23]), many non-pharmacological interventions, such as reality orientation, early mobility, and cognitive stimulation, have been shown to reduce the incidence and duration of delirium ([24]; [25]; [36]; [53]).

Cognitive stimulation (CS) engages patients in activities such as reality orientation, word searches, and board games to enhance cognitive functions, including attention, memory, and executive functions ([20]; [53]; [55]). Delivery methods range from traditional approaches like reality orientation ([48]; [52]) and workbooks ([44]) to brain-training applications and online games ([2]; [51]). In outpatient settings, CS typically involves 6 to 24 sessions, held 1–2 times weekly, each lasting 30 to 60 min ([10]). However, data on the duration, frequency, and implementation of CS therapy in acute hospital settings is limited. CS is increasingly recognized as a potential preventive intervention for delirium, primarily due to its ability to enhance cognitive reserve, promote neuroplasticity, improve sensory engagement, and facilitate social interaction. These mechanisms collectively support brain function and may help reduce delirium risk ([45]; [49]; [55]). However, implementing CS in acute hospital settings is challenged by factors such as limited resources, staffing shortages, and inadequate training among nursing staff ([46]). Additionally, traditional CS delivery often becomes repetitive, leading to diminished participant motivation and engagement ([28]). Recent studies suggest that incorporating games into CS and utilizing virtual reality (VR) for game-based CS can improve cognitive function ([53]), ([20]) and increase participant engagement ([5]; [8]; [21]; [31]; [34]; [35]). Thus, VR delivery of cognitive stimulation games is emerging as a potential solution to enhance patient engagement and overcome the scalability issues of current delirium prevention approaches.

This narrative literature review explores both traditional and novel digital-technology-based CS techniques for the prevention and management of delirium in acute hospital settings, focusing on research published in the past decade. The review expands upon prior key literature by specifically emphasizing CS as a non-pharmacological intervention for delirium prevention ([15]; [27]).

## 2. Methods

We undertook a narrative literature review of peer-reviewed articles from January 2014 to September 2024 to identify and critically analyze research on cognitive stimulation (CS) as a non-pharmacological therapy for preventing or managing delirium. The methodology for this review conformed to the Preferred Reporting Items for Systematic Reviews and Meta-Analysis (PRISMA) guidelines (Figure 1) ([33]). The population of interest included adults > 18 years old who were admitted to the hospital and had CS as a part of a CS exercise, cognitive pre-rehabilitation, or rehabilitation. We defined CS as any therapies or strategies directed at improving patient cognition or the domains of cognition. Examples of interventions included repeated tasks, games, skills, or questions, such as orientation exercises in both writing and/or verbal exercises delivered by healthcare professionals, family, computer software, or virtual reality. We sought to find studies comparing patients who received the intervention and those who did not and reported on our primary outcome of interest—delirium. The present narrative review included original research articles such as randomized controlled trials (RCT), quasi-experimental trials (i.e., non-RCT), observational trials, and pre/post-intervention trials describing the application of CS in the hospital setting and reporting of delirium according to validated tools such as the CAM-ICU ([38]). English-language publications were chosen to study adults in hospital settings.

### 2.1. Exclusion Criteria

The exclusion criteria were as follows: editorial, commentaries, abstracts, review articles, case reports, and letters with duplicate, incomplete, and unavailable data, as well as those studying participants with severe cognitive impairment, or with a history of severe sensory or motor impairment. For patients with terminal illnesses or a life expectancy of less than 6 months, where clinical outcomes may be heavily influenced by factors other than the intervention being studied, pharmacological interventions for the prevention or treatment of delirium were excluded. Non-English articles were excluded. Articles focusing only on CS following hospital discharge (i.e., outpatients) were excluded as we sought to assess interventions applied in hospital settings.

### 2.2. Search Strategy

H.F. developed search strategies and reviewed them with a health sciences librarian. The search was conducted using PubMed (n = 42), Scopus (n = 40), and Web of Science (n = 42) databases. Medical subject heading (MeSH) terms and keywords were used, including three key concepts:

“Delirium in the inpatient setting”, “ICU Delirium”, “Cognitive Stimulation”, and “Delirium in Older Adults”. Limitations included English-language articles. The initial search was conducted by the senior investigator (H.F.). Search results were managed using Covidence ([13]) and EndNote X9 software.

### 2.3. Screening Methods and Data Extraction

Three reviewers (E.F. and N.D.) manually screened duplicate titles and abstracts for predetermined inclusion and exclusion criteria. Titles and abstracts lacking sufficient information for inclusion were reviewed in full-text form. A librarian (A.T.) resolved disagreements. Articles were chosen for full-text review after assessment of inclusion criteria for the study population, study comparison, and study outcomes. Subsequently, two investigators (H.F. and I.H.) independently reviewed full-text articles for final data extraction and analysis.

### 2.4. Data Synthesis

The findings were presented in a narrative format. We performed a narrative literature review due to the heterogeneity of interventions, outcomes, and study designs.

## 3. Results

Figure 1 depicts the PRISMA flow diagram. The search identified 124 articles. After an initial review of the titles and research origin, 70 duplicate articles were eliminated, leaving 54 articles for further consideration. After the title and abstract review, 9 articles were removed for failure to meet inclusion criteria, leaving 43 publications for full-text review. After the full-text screening, 28 articles met the inclusion criteria. All 28 articles were reviewed using the standard extraction form, including the study sample, research methodology, research outcomes, and clinical implications, if available. Our literature search yielded twelve articles for in-depth analysis (Table 1). Table 2 outlines the types of CS protocols, including therapies, targeted cognitive domains, session frequency, duration, and therapy length. 

### 3.1. Healthcare Professional-Led Cognitive Stimulation

We identified eight studies ranging from feasibility studies to pre- and post-intervention studies, to randomized clinical trials in which healthcare professionals delivered CS, including nurses, occupational therapists, and physical therapists. There were wide variations in the type of outcome reported about delirium (e.g., incidence, duration, density, delirium-free days). A randomized clinical trial (RCT) by [9] ([9]) evaluated RAM-based CS therapy (CST) in 280 older patients with non-small cell lung cancer. Delirium screening was performed using the Nursing Delirium Screening Scale ([19]). The study found that the incidence of delirium was significantly lower in the CST group (10.71%) compared to the control group (20.71%). Faustino et al. ([18]) conducted an RCT with 144 critically ill patients to assess non-pharmacological interventions, including reorientation, CS, sensory correction, environmental management, and sleep promotion. Delirium incidence density was measured using the Confusion Assessment Method for the Intensive Care Unit (CAM-ICU) tool ([38]). The experimental group had a significantly lower incidence of delirium (1.3 × 10^−2^ person-days) compared to the control group (2.3 × 10^−2^ person-days), with a hazard ratio of 0.40 (95% CI: 0.17–0.95; *p* = 0.04). Felipe Martinez et al. ([37]) studied the impact of tailored interventions, including early mobilization, physical therapy, CS, and family involvement, on 227 adult ICU patients. Delirium was measured using the CAM-ICU tool ([38]). The study found a significant reduction in delirium incidence, decreasing from 38% to 24%, with a relative risk of 0.62 (*p* = 0.02). Mudge et al. ([41]) examined the impact of a structured early rehabilitation program, including early physiotherapy, an individualized exercise program, nursing support for functional independence, and CS activities on 124 patients aged 65 and older. Delirium was identified according to chart review using validated methodology ([26]). The author reported that the intervention group experienced a lower incidence of delirium than the control group (19.4% vs. 35.5%, *p* = 0.04). Álvarez et al. ([4]) conducted an RCT of 140 elderly ICU patients. Delirium screening was performed using the CAM-ICU ([38]). The study results showed a reduced incidence of delirium (20% in the control group vs. 3% in the experimental group) after implementing an occupational therapy-led cognitive intervention protocol that included stimulation, rehabilitation, and training exercises (*p* = 0.001). Rivosecchi et al. ([47]) evaluated CS as part of a non-pharmacological delirium prevention bundle that included nursing education, music, reorientation, and sensory care per pain, agitation, and delirium management guidelines ([6]). Intensive Care Delirium Screening Checklist (ICDSC) ([7]) was utilized to screen delirium. The author and their colleagues studied 230 patients in the pre-implementation phase and 253 in the post-implementation phase, reporting a decrease in delirium incidence from 15.7% in Phase I to 9.4% in Phase II (*p* = 0.04). Colombo et al. ([11]) reported a significant reduction in delirium occurrence, measured using CAM ([38]). The study results showed that delirium occurrence decreased from 36% in Phase I to 22% in Phase II following the introduction of a cognitive simulation protocol that included orientation, environmental, acoustic, and visual interventions (*p* = 0.020). This was controlled for dementia, APACHE II score, and mechanical ventilation. [50] ([50]) evaluated a two-stage intervention involving sensory stimulation and sleep hygiene in a pretest–posttest control trial with 92 COVID-19 ICU patients. Screening of delirium was performed using the CAM ICU tool ([38]). They found a significant reduction in delirium, with 56% of the experimental group being affected compared to 80% in the control group (*p* < 0.05).

### 3.2. Family Led Cognitive Stimulations

We found two studies in which a family member delivered CS. [39] ([39]) conducted a single-center randomized controlled trial with 90 patients, examining a family-delivered intervention that included daily orientation, sensory checks, and CS through discussions about family life and reminiscing. They reported the intervention as feasible and acceptable despite a low recruitment rate of 28%. [42] ([42]) conducted a three-arm RCT of 30 patients, testing a family-led intervention via voice recordings and found an increase in mean delirium-free days (evaluated by CAM-ICU) ([38]) in the family voice recording group (1.9 days) vs. the control group (1.6 days; *p* = 0.04).

### 3.3. Software-Based Cognitive Stimulation

E.A. Alvarez et al. ([3]) feasibility study evaluated software, including modules for time-spatial re-orientation, CS, early mobilization, sensorial support use promotion, sleep hygiene, and pain management optimization. The clinical feasibility assessment showed that 83.3% of the 30 enrolled hospitalized patients (76 ± 8 years) completed the 5-day protocol of software usage during hospitalization. Delirium was measured using the CAM-ICU tool ([38]). Software use was associated with a decrease in delirium incidence of 5 of 32 (15.6%) at baseline to 2 of 30 (6.6%) after its implementation.

### 3.4. Virtual Reality-Based Cognitive Stimulation

([16]) developed a prototype VR platform, “ReCognitionVR”, designed for immersive CS games. Initial testing included a 20 min VR session with healthy older volunteers ([16]), followed by a pilot trial with low-risk older surgical patients ([17]). Preliminary findings showed that the VR games were feasible, safe, and well-accepted, with all patients completing the sessions and achieving a mean System Usability Scale (SUS) score of 92 (SD = 8) without safety concerns. Game performance was assessed through metrics like the percentage of balloons popped and completion time, but no significant differences were found between groups. Due to the small sample size, the study did not observe any differences in delirium occurrence.

## 4. Discussion

Delirium emerges from a complex interaction among predisposing vulnerability risk factors, such as prior cognitive impairment, and acute insults, such as undergoing major surgery ([1]; [30]; [54]). Fortunately, delirium is preventable in up to 50% of patients, with the best preventive strategy being non-pharmacological interventions such as cognitive and sensory stimulation ([24]; [25]; [36]; [53]).

CS has been a well-established intervention in outpatient settings for managing cognitive impairment, dementia, and Alzheimer’s disease for decades ([20]; [55]). However, its application in inpatient and acute hospital settings remains limited. This is partly due to several challenges, including the complexity and high variability of delivery methods, which can hinder consistent implementation. Furthermore, low patient engagement and the existing constraints on the clinical workforce in hospital settings further exacerbate the difficulties in scaling these therapeutic approaches. These barriers highlight the need for more adaptable and resource-efficient models of CS that can be effectively integrated into acute care environments ([12]; [40]).

The Stress Recovery Theory and Attention Restoration Theory provide a framework for understanding how cognitive and sensory stimulation can mitigate ICU delirium ([43]). The Stress Recovery Theory emphasizes that exposure to natural environments promotes emotional and physiological recovery by activating the parasympathetic nervous system, especially in high-stress ICU environments ([43]). In contrast, the Attention Restoration Theory suggests that natural environments require less cognitive effort, facilitating disengagement from stress and restoring attentional resources ([43]). Together, these theories highlight the potential of novel digital-based CS interventions in alleviating the cognitive and emotional burdens associated with ICU settings. Therefore, this narrative literature review explores both traditional and novel digital-based CS techniques for the prevention and management of delirium in acute hospital settings. Considering the number of studies included in this paper, there is a clear need for additional research on applying CS therapy in the hospital setting.

We identified only ten studies that evaluated CS for delirium prevention but exhibited significant bias, warranting caution in their clinical application (Table 2). Healthcare professional-led CS demonstrated a variable reduction in delirium, as evidenced by three RCTs ([4]; [9]; [18]), three pre- and post-intervention studies ([37]; [47]; [50]), and one study using a chart-based identification method for delirium ([41]). One study focused solely on the feasibility of implementing a prevention program and did not report on delirium-related outcomes ([11]; [39]; [42]).

Family-led CS for hospitalized patients is under-utilized and under-reported, warranting further investigation. It presents an alternative to healthcare professional-led CS, which faces challenges such as limited resources, staffing issues, and insufficient training among nursing staff ([46]). Feasibility studies on family-led CS included in this review ([39]; [42]), while not sufficiently powered to assess delirium outcomes, can aid in developing protocols for future randomized controlled trials. For instance, Mitchell et al. provided a sample size estimate of 596 for achieving 80% power at a significance level of (*p* = 0.05) ([39]).

Conventional strategies used for CS in the ambulatory setting range from traditional reality orientation ([48]; [52]) and workbooks ([44]) to brain-training applications and online games ([2]; [51]). Healthcare professionals and family-led conventional CS can often become repetitive, decreasing motivation and participant disengagement ([28]). Incorporating games into CS therapy and using virtual reality (VR) to deliver such cognitive games has been shown to enhance cognitive functions ([53]; [20]) and increase participant engagement ([5]; [8]; [21]; [31]; [34]; [35]). Thus, VR delivery of CS games is emerging as a potential solution to enhance patient engagement and overcome the scalability issues of the current delirium prevention approaches. This review identified one study on software-based CS and two studies on VR-based CS, all exhibiting small sample sizes and a high risk of bias. Consequently, the efficacy of these interventions for delirium prevention and management remains inconclusive. In a feasibility study by [3] ([3]) software-based re-orientation and CS correlated with a reduction in delirium incidence from 15.6% (5 of 32) at baseline to 6.6% (2 of 30) post-implementation. [17] ([17]) assessed VR-based CS games for older surgical patients, finding them safe, feasible, and acceptable, though delirium outcomes were not reported.

The variation in session frequency, length, and intervention duration across studies restricts the generalizability of CS protocols in acute hospital settings ([4]; [9]; [41]; [47]). Additionally, many of the studies we reviewed had small sample sizes or were feasibility studies with a high risk of bias ([3]; [17]; [39]; [42]). As a result, it is premature to draw firm conclusions regarding the specific types, frequencies, durations, and components of CS that would be effective in preventing or reducing the duration of delirium in this population. Further well-designed, large-scale studies are needed to establish more robust evidence on the optimal characteristics of CS protocols in acute care hospital settings.

### Limitations

This narrative review has several limitations. Firstly, it focused exclusively on peer-reviewed literature published in English, potentially omitting relevant studies in other languages and unpublished work. Secondly, it excluded other reviews, case reports, and commentaries. Thirdly, the included studies were assessed to have a critical, serious, or high risk of bias, which restricts the ability to draw definitive conclusions regarding the effects of CS. Fourthly, most of the studies were pilot or feasibility studies, indicating that it is too soon to determine their impacts on delirium outcomes. Lastly, the review may be constrained by the specific databases searched, with potentially relevant studies not indexed in these sources.

## 5. Conclusions

CS for delirium prevention in a hospital setting is a relatively new area of research and warrants further exploration. In addition, the implementation of CS in hospital settings faces several challenges. VR delivery of CS games is emerging as a potential solution to enhance patient engagement and overcome the scalability issues of current delirium prevention approaches. However, insufficient supporting evidence is available to recommend its use. As such, the authors are conducting a study to evaluate VR-based CS games to prevent delirium in older adults in a hospital setting. Larger, multi-center trials to evaluate VR-based cognitive intervention protocols are needed to examine the effects on delirium outcomes in a hospital setting.

## Figures and Tables

**Figure 1 behavsci-15-00186-f001:**
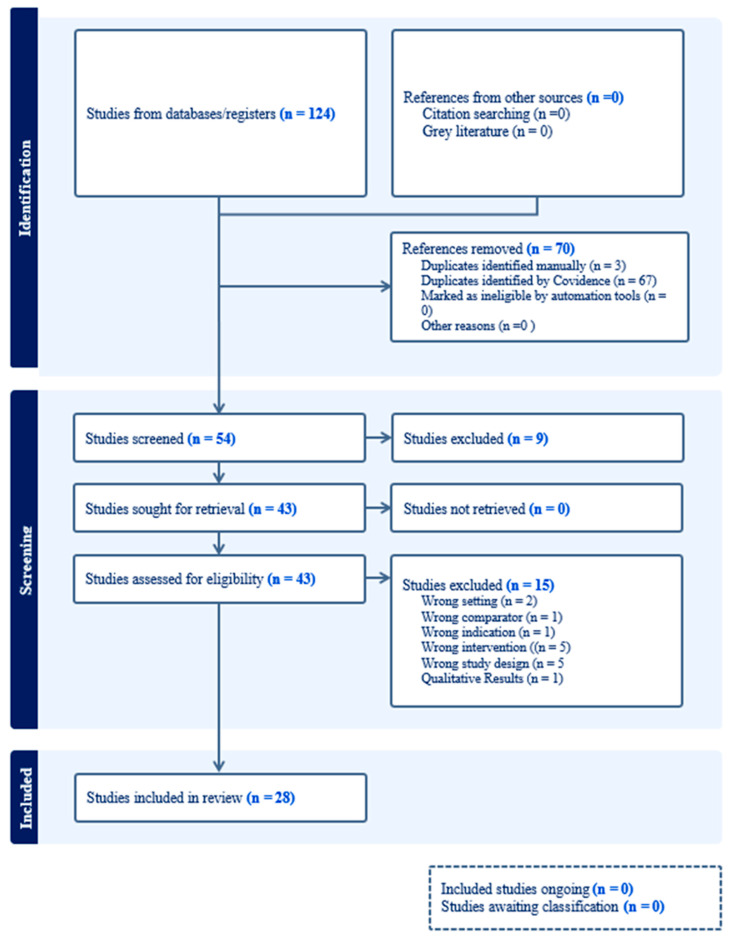
PRISMA flowchart of study selection.

**Table 1 behavsci-15-00186-t001:** Study Summary.

Study	Purpose/Intervention	Design, Age, Sample Size	Outcomes and Outcome Measures	Results	Conclusion
**Healthcare Professional-led cognitive stimulation**					
[9] ([9])	To investigate the effects of a Royal Adaptation Model (RAM)-based cognitive stimulation therapy (CST) on older patients with primary non-small cell lung cancer (NSCLC) undergoing curative resection	Single-center randomized controlled trial (RCT) Age > 65 years n = 280	Delirium prevalence/incidence using the Nursing Delirium Screening Scale.	Incidence of delirium: 20.71% in the control group vs. 10.71% in the RAM-based CST group (*p* = 0.032)	RAM-based CST in elderly NSCLC patients undergoing curative resection yielded reduced delirium incidence.
[18] ([18])	To evaluate the effectiveness of combined non-pharmacological interventions (periodic reorientation, cognitive stimulation, correction of sensory deficits) in preventing delirium in critically ill patients	Single-center RCT Age > 18 years n = 144	Delirium incidence density using the Confusion Assessment Method for the Intensive Care Unit (CAM-ICU) tool.	Incidence density of delirium: (2.3 × 10^−2^ person-days) in control group vs. (1.3 × 10^−2^ person-days) in the intervention group.	Combined non-pharmacological interventions reduced delirium in critically ill patients compared to standard care.
[37] ([37])	To assess the effectiveness of a tailored multicomponent intervention (early mobilization for preventing the incidence of delirium among critically ill patients.	Before-and-after study Age > 18 years n = 227	Delirium incidence using CAM-ICU tool.	Incidence of delirium: Reduced from 38% to 24% (relative risk, 0.62; 95% CI, 0.40–0.94; *p* = 0.02)	Multicomponent strategy successfully reduced delirium. Early participation of the whole team, shared leadership, and the provision of concrete tasks were key to the intervention’s success.
[41] ([41])	To evaluate the effect of a structured, multi-component, early rehabilitation program on delirium of older acute medical inpatients.	Prospective controlled trial Age ≥ 65 years n = 124	Incidence of delirium. Delirium was identified according to chart review using validated methodology.	Incidence of delirium: 35.5% in control group vs. 19.4% in the intervention group (*p* = 0.19)	In the intervention group, there was a reduction in delirium.
[4] ([4])	To determine the impact of occupational therapy (OT)-led cognitive intervention protocol on the incidence, duration, and severity of delirium in older ICU patients	Pilot study, RCT Age > 60 years n = 140	Delirium incidence and duration using the CAM ICU tool.	Incidence of delirium: 20% in the control group vs. 3% in the treatment group (*p* = 0.01) Duration of delirium: lower in the treatment group (IRR, 0.15; 95% CI, 0.12 to 0.19; *p* < 0.001): Control group (IRR, 6.7; 95% CI, 5.2 to 8.3; *p* < 0.001).	A combination of early OT and cognitive intervention strategies decreases the incidence and duration of delirium.
[47] ([47])	To assess whether an evidence-based non-pharmacologic protocol could further decrease the duration of delirium in patients in a medical ICU that already implements a sedation and mobility protocol.	Prospective, pre–post intervention QI project. (n = 483). Phase I: baseline data collection before protocol implementation (n = 230). Phase II: development and implementation of non-pharmacologic protocol	Incidence and duration of delirium in phase 1 vs. 2, using the Intensive Care Delirium Screening Checklist (ICDSC).	Phase I vs. Phase II delirium incidence (15.7% vs. 9.4%; *p* = 0.04). Median duration of delirium in Phase I (20 h) and Phase II (16 h), (50.6% reduction; *p* < 0.001)	Non-pharmacologic strategies reduce risk and duration of delirium in the ICU, even if a mobilization protocol and sedation algorithm are already in place.
[11] ([11])	To assess the efficacy of the cognitive stimulation protocol (orientation, environmental, acoustic, and visual interventions) on delirium in medical and surgical ICU patients	Two-stage prospective-observational study. Age > 18 years Phase 1: observational (n = 170) phase II interventional (n = 144)	Delirium occurrence using CAM-ICU tool.	Delirium occurrence was lower (36% in phase I vs. 22% in phase II, *p* = 0.02).	A reorientation strategy was associated with a reduced incidence of delirium.
[50] ([50])	To evaluate the effect of two-stage intervention (sensory stimulation and sleep hygiene) on delirium in Coronavirus disease-2019 (COVID-19) patients	Pre-test/post-test control group and trial model. Age > 18 years n = 92	Delirium incidence using CAM-ICU tool.	Incidence of delirium: 80% in control group vs. 56% in the intervention group (*p* < 0.05)	The sensory stimulation and sleep hygiene intervention based on the nursing model effectively reduced the incidence of delirium in critically ill COVID-19 patients.
**Family-led cognitive stimulations**					
[39] ([39])	To evaluate the feasibility and acceptability of a family-delivered intervention (orientation or memory clues, sensory checks, and therapeutic or cognitive stimulation) to reduce delirium in hospitalized ICU patients.	Single-center feasibility RCT Age ≥ 16 years n = 61	Retention of family members, feasibility, and acceptability of the intervention	No family member withdrew from the intervention group, and one withdrew from the control group. Low recruitment rate (28%)	The feasibility of recruiting and retaining family member participants; nurses supportive of interventions
[42] ([42])	To determine if recorded audio-orienting messages (automated orientation messages in a family member’s voice) reduce the risk of delirium in critically ill adults.	Prospective RCT Age > 18 years n = 30	Delirium-free days, as evaluated by CAM-ICU.	Mean delirium-free days: 1.9 in the family voice group, 1.6unknownvogroup, and 1.6 in the control group (*p* =0.04)	Participants exposed to recorded voice messages from family members had more delirium-free days.
**Software-based and Virtual-Reality (VR)-based cognitive stimulation**					
[3] ([3])	To determine the clinical feasibility assessment of software by older adults	Feasibility study Age > 75 years n = 30	Delirium incidence using the CAM-ICU tool.	Software use was associated with a decrease in delirium incidence of 5 of 32 (15.6%) at baseline to 2 of 30 (6.6%) after its implementation.	Use of software to improve the delivery of non-pharmacological interventions may prevent delirium.
[17] ([17])	To determine VR-based cognitive stimulation games’ safety, feasibility, and acceptability for preventing delirium in older surgical patients.	Pilot trial Age ≥ 60 years n = 30	Safety, feasibility, and acceptability. Delirium incidence using the CAM tool.	ReCognitionVR-based cognitive games were safe, feasible, and meaningful Mean System Usability Scale (SUS) score of 92 (SD = 8)	The study did not observe any differences in delirium occurrence due to the small sample size.

**Table 2 behavsci-15-00186-t002:** Summary of cognitive stimulation interventions.

Study	Specific Cognitive Domains Targeted	Specific Therapies	Session’s Frequency and Length, and the Total Duration of Therapies
**Healthcare Professional-led cognitive stimulation**			
[9] ([9])	Attention, concentration, and memory	Memory and Attention Training: Conducting various activities twice weekly within the first month after surgery, aimed at enhancing memory and attention by evoking childhood memories, and current events, with sounds and word games, and with each session lasting one hour	One-hour sessions, twice a week, during the first month after surgery.
[18] ([18])	Attention, memory, and executive function	Periodic reorientation twice a day Digit span, digit game, memory task, block test, executive functioning, bells test, and difference searching	Not available
[37] ([37])	Orientation	Reorientation on a date, time of day, and location at least twice a day Explain the reason for admission to the unit to all patients daily	Not available
[41] ([41])	Orientation and memory	Socialization, orientation, and memory activities Discussion of approaches to psychological aspects of hospitalization, especially anxiety and depression	Sessions offered 3 to 4 afternoons per week.
[4] ([4])	Alertness, visual perception, memory, calculus, problem-solving, praxis, and language	Poly-sensory stimulation (intense external stimuli). Notebooks, sequencing cards, and games, e.g., dominoes, playing cards, memory, and visuospatial construction	Two 40 min sessions per day, one in the morning and one in the evening, for 5 consecutive days.
[47] ([47])	Orientation	Reorientation Cognitive stimulation questions Music therapy Television Hearing aids and glasses	Not available
[11] ([11])	Not identified	Five W’s and one H Scale: (Who are you and who is the nurse/physician? What happened?; When did it happen and what is the date?; Where are you/we?; Why did it happen?; How did it happen and what is the illness progression?) Mnemonic stimulation (i.e., remembering relatives names). Environmental, acoustic, and visual stimulation (i.e., wall clock, reading of newspapers/books, listening to music/radio)	Not available
[50] ([50])	Not identified	Sensory stimulation includes reorientation	Not available
**Family-led cognitive stimulations**			
[39] ([39])	Orientation	Whiteboard day planner Family photographs Family orientation of patient Family discussion of personal events and patient interests Family ensuring appropriate sensory aids (i.e., glasses, hearing aids)	Once per day during at least four ICU visits
[42] ([42])	Orientation	Audio recording of orientation message	2 min long hourly messages, 8 per day, for 3 days.
**Software-based and Virtual-Reality (VR)-based cognitive stimulation**			
[3] ([3])	Attention and memory	A desktop module continuously delivering information on time and space to facilitate orientation and allow access to other modules. A games module including eight different cognitive activities to stimulate attention, memory, and executive function	The software was installed on a tablet, an Alcatel OneTouch Pixi-3, and delivered daily to patients between 09:00 and 20:00.
[17] ([17])	Attention, executive function, and memory	VR-based low-cognitive exercises, including a relaxed environment, time orientation, delivery instructions, and task completion motivation	20 min for one session

## Data Availability

No new data were created or analyzed in this study.

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
