# Peer review of "Exploring Cognitive Stimulation as a Therapy for the Prevention of Delirium in a Hospital Setting: A Narrative Review"

_behavsci, 2025, doi:10.3390/bs15020186_

Round 1

Reviewer 1 Report

Comments and Suggestions for Authors

The aim is to review the research concerning the cognitive stimulation of delirium patients. In the reviewer's opinion, the real aim was not so much to review the literature for the purpose indicated in the abstract, but rather to justify the lack of research on this issue AND the superiority of VR applications over other techniques. This is evident not only in the individual steps of the analyses, but also in the fact that the authors point out the limitations of all methods except VR.
The theoretical background lacks an explanation of the concept of delirium and its symptomatology. It would also be desirable to point out the link between individual symptoms and cognitive areas. This is important in the context of the cognitive nature of the literature review.
The research methodology is not objectionable. I propose to complete the theoretical basis and to clarify the actual purpose of the literature review.

Author Response

Reviewer 1:

 Comments and Suggestions for Author

 Comments: The aim is to review the research concerning the cognitive stimulation of delirium patients. In the reviewer's opinion, the real aim was not so much to review the literature for the purpose indicated in the abstract, but rather to justify the lack of research on this issue AND the superiority of VR applications over other techniques. This is evident not only in the individual steps of the analyses, but also in the fact that the authors point out the limitations of all methods except VR.
The theoretical background lacks an explanation of the concept of delirium and its symptomatology. It would also be desirable to point out the link between individual symptoms and cognitive areas. This is important in the context of the cognitive nature of the literature review.

The research methodology is not objectionable. I propose to complete the theoretical basis and to clarify the actual purpose of the literature review.

Author Response:  Your comments are appreciated.  The abstract has been revised with an explanation of delirium symptomatology. (Abstract, page # 1; Introduction, page #3)

The link between individual symptoms of delirium and cognitive areas has been explained in detail. (Introduction, page # 3)

The purpose of the literature review is explained with clarity. (Abstract, page # 1; Introduction, page #4)

Reviewer 2:

The authors have submitted an interesting narrative review of the use of cognitive stimulation approaches for prevention of delirium in the hospital. As the authors state, this is an understudied area of research within the field of delirium and I appreciate their efforts in providing a succinct review on this to educate and bring light to this topic. There are not very many studies as cited by the authors, but RCTs have been conducted, thus providing reinforcement to the idea that this should be investigated further. The overall scope and premise is very good, and I have a few thoughts to help improve the message of the paper for the authors to consider:

Author response: We appreciate your comments

Comment 1: I would recommend being clear on the populations being studied by the RCTs and the type of delirium being discussed by the authors. The abstract and introduction focus on postoperative delirium; however a fair amount of the studies cited actually involve delirium in the ICU. This is important as the etiology of the delirium may lead to different outcomes and different pathophysiology involved (e.g., surgery-related causes causing delirium vs multi-focal pneumonia + a urinary tract infection and sepsis in the ICU). Otherwise this can be confusing for the reader. 

Author response: We appreciate your suggestion and have clarified the term delirium. Although most studies focus on ICU patients, some have included patients on floors and wards. ICU delirium may include both medical and surgical patients. (Methods, page # 6)

Comment 2: I would suggest summarizing and explaining cognitive stimulation further and what this encompasses and how it is carried out. I would even consider creating a separate table to make it clear what these protocols entailed in the studies cited as cognitive stimulation is a broad term and can involve many exercises and protocols. This would enhance clarity and help the reader understand how this can be carried out and how we can improve upon it further (e.g., how many times, how long each session, how many exercises, how many cognitive domains).

Author response: We appreciate the above suggestion. We have explained cognitive stimulation in detail and created a separate Table 2 on the cognitive stimulation protocols. (Table 2; Results, page #7)

Comment 3: I would suggest some thoughts as to why CS can prevent delirium in specific settings; what about CS as an intervention tied to the neurobiology and pathophysiology of delirium development? 

Author response: We have explained the theoretical background and the link between how CS can prevent delirium. ( Discussion, page # 10)

Reviewer 2 Report

Comments and Suggestions for Authors

The authors have submitted an interesting narrative review of the use of cognitive stimulation approaches for prevention of delirium in the hospital. As the authors state, this is an understudied area of research within the field of delirium and I appreciate their efforts in providing a succinct review on this to educate and bring light to this topic. There are not very many studies as cited by the authors, but RCTs have been conducted, thus providing reinforcement to the idea that this should be investigated further. The overall scope and premise is very good, and I have a few thoughts to help improve the message of the paper for the authors to consider:

1. I would recommend being clear on the populations being studied by the RCTs and the type of delirium being discussed by the authors. The abstract and introduction focus on postoperative delirium; however a fair amount of the studies cited actually involve delirium in the ICU. This is important as the etiology of the delirium may lead to different outcomes and different pathophysiology involved (e.g., surgery-related causes causing delirium vs multi-focal pneumonia + a urinary tract infection and sepsis in the ICU). Otherwise this can be confusing for the reader. 

2. I would suggest summarizing and explaining cognitive stimulation further and what this encompasses and how it is carried out. I would even consider creating a separate table to make it clear what these protocols entailed in the studies cited as cognitive stimulation is a broad term and can involve many exercises and protocols. This would enhance clarity and help the reader understand how this can be carried out and how we can improve upon it further (e.g., how many times, how long each session, how many exercises, how many cognitive domains)

3. I would suggest some thought as to why CS can prevent delirium in specific settings, what about CS as an intervention ties to the neurobiology and pathophysiology of delirium development? 

Thank you for your hard work. 

Author Response

Please attached document

Round 2

Reviewer 1 Report

Comments and Suggestions for Authors

Thank you for completing the manuscript as suggested.